# Description of the Gap between Local Agricultural Practices and Agroecological Soil Management Tools in Zerhoun and in the Middle Atlas Areas of Morocco

Aziza Irhza [1,2], Laila Nassiri [1], Moussa El Jarroudi [3,*], Fouad Rachidi [1], Rachid Lahlali [1] and Ghizlane Echchgadda [1]

1 Plant Ecology Unit, Department of Plant Protection and Environment, National School of Agriculture, Meknes 50001, Morocco
2 Environment and Valorisation of Plant and Microbial Resources Unit, Department of Biology, Faculty of Sciences, Moulay Ismail University, Meknes 11201, Morocco
3 Department of Environmental Sciences and Management, SPHERES Research Unit, University of Liège, 6700 Arlon, Belgium
* Correspondence: meljarroudi@uliege.be

**Abstract:** Agroecology is an agricultural, holistic, and innovative approach, which aims to increase the effectiveness, efficiency, and sustainability of agricultural systems. This approach is very rewarding for agriculture in developing countries, mainly in arid zones where water resources are in sharp decline. In this study, we examined farmers' knowledge by studying existing agricultural production systems and agricultural practices in Moulay Driss Zerhoun (S1) and the Middle Atlas (S2) in central Morocco, to assess the gap between them and the principles and recommendations of agroecology. Data were collected through a survey of 64 farmers, and through field trips and observations. Collected data were analyzed with Chi-square tests and canonical correspondence analysis. Most farms (63.3% in S1 and 52.9% in S2) have an area between 0 and 5 ha. Both areas frequently opt for agroforestry. Olive trees and annual crops (85%), olive trees and fodder crops (10%), and olive trees and market garden crops (5%) are the most recorded associations. Olive trees were used frequently in association with other fruit trees, such as almond trees (15%), carob trees (13%), and fig trees (5%). Of farmers practicing agroforestry, 70% use only organic fertilizers. Thus, 53% of the farmers interviewed at the level of the S2 zone cultivate vegetable crops, against 17% at the level of S1, and only 40% of the farmers of S1 use irrigation, while 60% depend on rainfall. On the other hand, 83.3% of farmers in S1 adopted a two-year rotation—cereals and legumes—against 30% in S2. Tillage or plowing is considered by farmers a necessary procedure before sowing, and, only in S2, 71% of farmers opt for annual tillage of their land for agroforestry. Based on multivariate analysis, the choice of crops was significantly influenced by region and type of organization. However, it was not influenced by topography or farmland size, and the land and area played an important role in the selection of crop types. Raising awareness and introducing agroecological practices in the two study areas based on local knowledge seems essential, with the aim of strengthening the resilience of agroecosystems, respecting the environment, and guaranteeing the sustainability of small farmers through the diversification of their productions. The obtained results from this work are the first in this field of study and constitute a basis for comparative investigations.

**Keywords:** agroecology; agricultural practices; small producers; production systems

## 1. Introduction

Due to climate change, agricultural production is anticipated to decrease, especially in desert regions [1]. These areas are known for their arid climate, low rainfall, and scarcity of fertile soils, which affect negatively agriculture. Due to low precipitation, the arid environment impacts agronomy fields through the reduction of irrigation water, while increased temperatures negatively affect the growth, physiological traits, and productivity

of crops [2,3]. Therefore, traditional agriculture is not appropriate for these environments and new innovative and sustainable approaches, such as agroecology, are needed to help agrosystems continue their production in arid areas.

Morocco's population will be particularly susceptible to food insecurity in the future due to the country's growing population and rising food demand [4,5]. Forecasts show that among Middle Eastern and North African nations, Morocco will face the biggest imbalance in precipitation caused by climate change, with rising annual average temperatures, declining precipitation, declining groundwater recharge, and rising water stress [6–8]. Wheat production, a key crop for Moroccans, is highly correlated with annual precipitation [9,10]. Additionally, because of its large reliance on imports and openness to the outside world, the nation is extremely vulnerable to exogenous shocks, such as inflation in food prices [11]. Therefore, the installation of sustainable modes of agriculture is suggested to maintain productivity. However, before the implementation of a new agricultural approach, analysis of the existing production systems and applied practices is of great importance.

The agricultural sector in Morocco is characterized by a dualism; intensive agricultural production in irrigated farms that represent only a small fraction of cultivated land [12], and which contrast sharply with the dominant traditional agricultural system which is subsistence-oriented, largely rain-fed and employs mostly traditional farming practices [13]. Therefore, it has become essential to adopt an innovative and efficient approach to agriculture to enable agriculture to meet the challenge of food security in a manner that is efficient (economically and socially) and respectful of the environment [14]. However, the evaluation of existing practices, soil fertility, and irrigation approaches is suggested to permit the evaluation of their capacity to deal with new challenges, such as climate change and scarcity of water.

In Morocco, the agroecology is gaining interest among decision makers as a possible answer to the challenges that agricultural systems are facing [15–17]. Therefore, this paper investigates three aspects in relation to agricultural systems and agroecology basis on two study sites, Moulay Driss Zerhoun and the Middle Atlas, located in the Northern and Southern limits of the Saiss plain, respectively. First, this study aimed to characterize farmers and describe the agricultural systems in the selected sites. Second, we sought to analyze the relationship between agricultural practices and crop types among participating farmers. Third, we analyzed the local agricultural production system regarding the agroecology basis. The study zones were selected in the Saiss agricultural plain, considered one of the most fertile and productive regions in Morocco. Additionally, this plain is dominated by small farms [18]. Therefore, this area is of great importance in analysing the relationship between agricultural practices and agroecology principles. Our hypothesis is that the agricultural systems in the Saiss plain are practicing local activities far from agroecology recommendations in terms of soil fertilization, irrigation, and selection of cultivated seeds.

## 2. Materials and Methods

### 2.1. Study Area

This study was conducted in two rural regions of Morocco from December 2020 to March 2021 (Figure 1). The first site (S1), Zerhoun (34°3′10.25″ N, 5°30′2.57″ W), is located in the Zerhoun massif within the Saiss region. The Zerhoun massif is located in one of Morocco's most favorable cropping regions, with a typical Mediterranean climate [15,19]. This agricultural plain is considered one of the most fertile and productive regions in Morocco, and in addition, this plain is dominated by small farms. Ninety percent of rainfall occurs between November and April, with a mean annual precipitation of 580 mm. The mean annual minimum and maximum temperatures are 11 °C and 28 °C, respectively [15]. Elevations across the massif range from below 300 m in the fertile plain west of Moulay Idriss Zerhoun, rising to above 1000 m at the peak of Jbel Zerhoun [20].

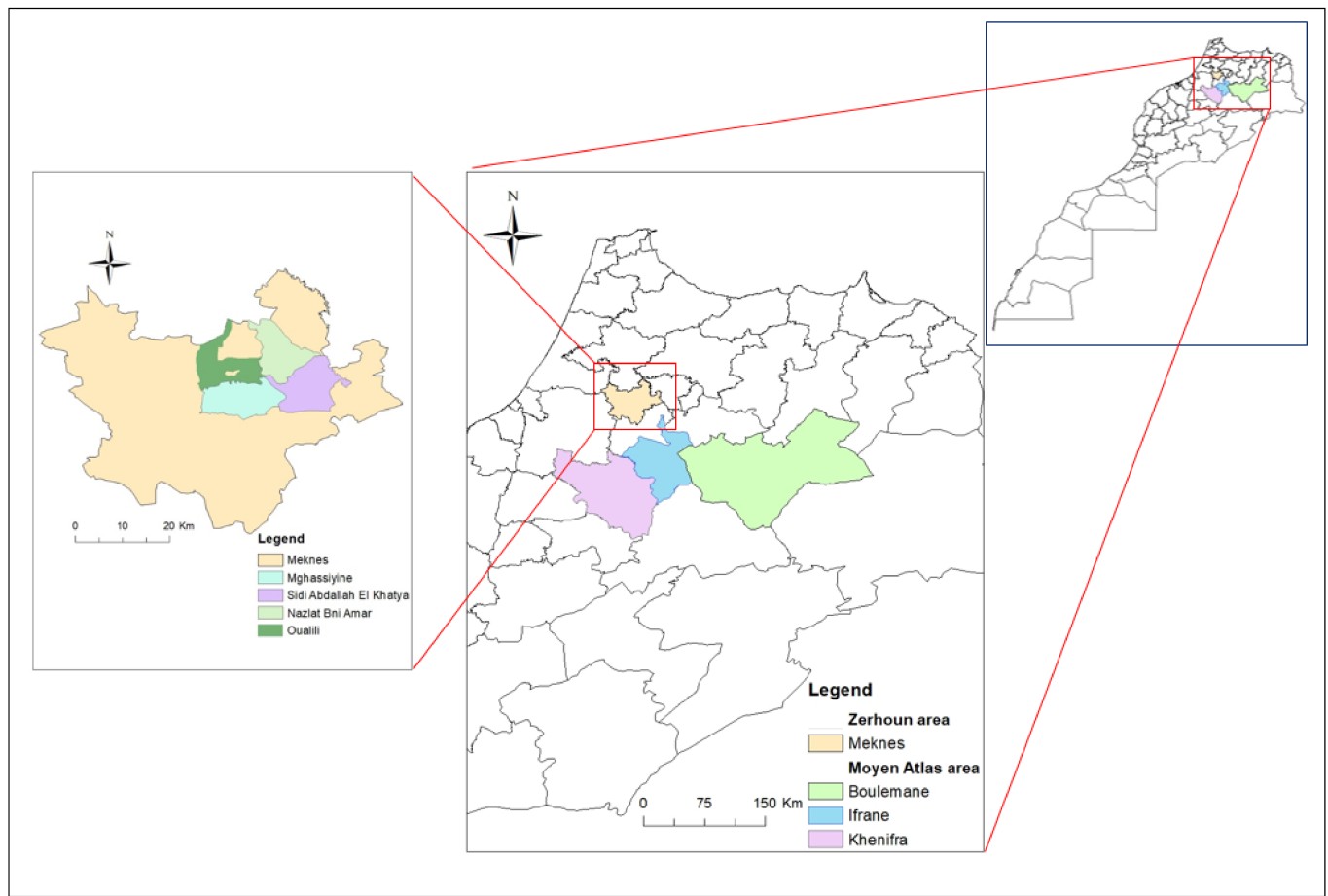

**Figure 1.** Geographical location of study areas Zerhoun (S1) and Middle Atlas (S2).

Great proportions of farmland are located on slopes greater than 15%. Soil types are varied, but calcareous vertisols dominate [21]. The field survey was conducted with farmers of four rural communes: Nzalet de Beni Amar, Kermet Ben Salem, El Mghassiyine and Walili. These administrative units surround Moulay Idriss Zerhoun urban areas. Rain-fed cereal production is the dominant land use. Legumes, olive (*Olea europaea*) and fruit trees are other crops cultivated at substantial scale [22,23]. Nationally, the site is famous for its olive groves and oil [22]. Local forest cover is extremely sparse [19]. Limited market access, adoption of suitable technological packages, land tenure, illiteracy and land fragmentation impede income generating from agriculture [15].

The second site (S2), Middle Atlas (33°26'58.98" N, 5°13'9.48" W), is located in the central Morocco region, characterized by a diverse climate from semi-arid to sub-humid, with an amount of rainfall between 400 and 1100 mm per year and a diversified vegetation cover [24,25]. The underground and surface water resources in this area are important [26]. The consequences of the climatic changes lie in the reduction in river flows in the Middle Atlas by around 20 to 30% [27,28]. In the Middle Atlas, agricultural land covers only 20%, and this low rate is explained by the fact that the local economy was once linked to rangeland [24]. Currently, it is based on a little-developed agriculture associated with extensive livestock production and forest exploitation (State domain), which occupies 24% of the space against 56% for extra-forest rangelands [29]. Topographic, soil and climatic constraints weigh more heavily on agricultural life than the population. The rural population density in S2 is around 20 inhabitants/km² on average and can reach 150 inhabitants/km² in the valleys where the possibilities of land reclamation are important.

*2.2. Sampling Method*

The multi-stage stratified random sampling method was used to select farms (preliminary step). In the first stage, all the provinces (Meknes, Ifrane, Khenifra, Boulemane) were selected. In the second stage, three townships were randomly selected in each province. In the third stage, we selected three villages in each township and contacted local village leaders to obtain help and basic information. Then, 5–8 farms were randomly selected from each village, irrespective of their land size or location of the farm [24].

*2.3. Survey and Data Collection*

A questionnaire was created and assembled to ask questions on the farming practices of the mainly small-scale farmers in both sites. The primary objective of the survey was to collect data on the agricultural production systems in both sites and to catalogue the agricultural production practices associated with each system (Tables A1 and A2 in Appendix A). A structured questionnaire was designed to explore our research objectives. The questionnaire was developed and revised based on a series of pretest surveys [30]. The final questionnaire used in the field contained four sections addressing the main topics: (1) farm characteristics; (2) agricultural practices of agroforestry; (3) agricultural practices of vegetable crops; and (4) agricultural practices of cereals and legumes. To more accurately evaluate the data gathered, the interviews were conducted outside in the farmers' fields. Surveys of farmers from various elevations were conducted to identify any potential variations in farming systems and practices.

*2.4. Data Analysis*

The raw data collected was examined, coded, and recorded into an Excel file at the conclusion of the survey. The relative proportions in each site were the basis for the statistical analysis, which was done using Statistical Package for the Social Sciences software (SPSS) (version 25, IBM SPSS Statistics 25, New York, NY, USA). First, we calculated the means and SD, then data was tested for normality via chapiro test. The % of each answer was (type of answers/total answers * 100). To compare the answers of participants between study sites, Chi-square test was employed ($p = 0.05$). (12). A map showing the location of the study area was prepared using ArcGIS software 10.3.1 To test the relationship between the farms' features and agricultural methods, we used canonical correspondence analysis, in which farms' features were considered dependent variables, while agricultural methods were considered factors (explanatory variables). Then test was conducted in XLStat [31].

**3. Results**

*3.1. Farms Characteristics*

In total, 64 farmers were questioned in both study areas. The 26 villages in the two research areas that these farmers are from were surveyed during the study period (Table 1). S2 (34 farmers) had more interviews with farmers than S1 (30 farmers). Since there were no female farmers in the fields at the time of the study, only male farmers were contacted. In S2, all participants are farmers, as opposed to S1, when 10% of the interviews were located in Moulay Driss Zerhoun's urban region.

Every farmer surveyed had proof of their tenancy on the land, since the farms at both locations are small. Renting of agricultural land was not common in either study site. Joint land ownership among family members was a common practice in the areas. The majority of farms at both locations are between 0 and 5 ha, making up 63.3% of S1 and 52.9% of S2, respectively. S2 generally has a larger farm than S1, with farms larger than 5 ha accounting for 47% and 36% of the total farm area, respectively. Farms in both locations are tiny in size and have numerous parcels.

Due to the differences in topography between the two sites, 47% of the farms in S2 and only 13% in S1 are situated on flat land, respectively. This indicates that 87% of the farms surveyed are situated entirely or in part on muddy terrain in S1, compared to 53% in S2. On the other hand, 90% of farmers in S1 have planted vegetation (often cactus) to

contour their parcels or preserved natural hedges for land conservation purposes. Hedge plants or natural ones are only present in 64% of the farms surveyed in S2.

**Table 1.** Identification of farms.

| Categories | Variables | Zerhoun | | Middle Atlas | |
|---|---|---|---|---|---|
| | | **n** | **%** | **N** | **%** |
| Commune | Urban | 3 | 10.0 | 0 | 0 |
| - | Rural | 27 | 90.0 | 34 | 100 |
| | | | | | |
| Type of professional | Cooperative | 2 | 6.7 | 0 | 0 |
| organization | Association | 3 | 10 | 7 | 20.6 |
| $X^2 = 0.869$ | Any organization | 25 | 83.3 | 27 | 79.4 |
| | | | | | |
| Farmland areas | 0–5 ha | | | 18 | 52.9 |
| $X^2 = 6.75$ | 5–20 ha | 19 | 63.3 | 12 | 35.3 |
| | >20 ha | 8 | 26.7 | 4 | 11.8 |
| | | 3 | 10.0 | | |
| | | | | | |
| Topography of farms | Flat | | | 16 | 47.1 |
| $X^2 = 9.719$ | Slope | | | 1 | 2.9 |
| | Both | | | 17 | 50.0 |
| | | 4 | 13.3 | | |
| Methods of fences | Living fences | 4 | 13.3 | 1 | 2.9 |
| $X^2 = 10.564$ | Artificial fences | 22 | 73.3 | 21 | 61.8 |
| | Any fences | | | 12 | 35.3 |
| | | | | | |
| | | 17 | 56.7 | | |
| | | 10 | 33.3 | | |
| | | 3 | 10 | | |

Every farmer examined grows annual crops (such as cereals, legumes, forages and vegetables) and has fruit trees (including olive, fig, carob, and almond trees). However, the final destination of agricultural production is either self-consumption or the local market, depending on the farm size, the degree of intensification, the amount produced, and the relevance of the crop produced.

The majority of respondents depend on rain. Nevertheless, access to moto-pumps allows a few lowland farmers in S2 to do supplemental irrigation on small sections of their cropland, through traditional flooding or drip irrigation.

Both zones frequently utilize agroforestry. S1 is dominated by olive, carob, fig, and caper shrubs, whereas S2 is dominated by fruit trees (apples, cherries, peaches, prunes, pears, etc.). Only 5% of farmers in S1 grow vegetables for their personal consumption in our situation. Olive trees and annual crops (85%), olive trees and fodder (10%), and olive trees and market gardening (5%) are all connections that coexist. Olive trees frequently grow in associations with other fruit trees, such as olive trees and almond trees (15%), olive trees and carob trees (13%), and olive trees and fig trees (5%), especially on plots with few olive trees and on steep slopes. In contrast, more than half (53%) of the farmers surveyed grow vegetables for both local markets and their own consumption in S2. As for cereals (dominated by barley) and legumes (dominated by faba beans), almost 87% of participants farmers in S1 grow them because they are the basis for their family meals and diet, compared to 59% in S2. On the other hand, seventy percent of farmers interviewed in S1 integrate livestock in their agricultural production system, compared to almost 62% in S2. In both study areas, most livestock owners have very small land properties and rely on rangeland to feed their herds.

*3.2. Description of the Agricultural Production Systems*

Agroforestry, vegetable production, and grains and legumes production are the three agricultural production techniques used by farmers in sites S1 and S2, according to surveys. This high productivity is made possible by the abundance of flat lands, fertile soil, and water, all of which are essential for agriculture and agroforestry.

3.2.1. Agroforestry Agricultural Practices

If local conditions are adequate, one of the adaptation strategies based on trees inter-cropping that would be a fundamental resilience strategy for small producers is agroforestry. Regardless of strata, agroforestry activities were present in all farming systems. The extent of tree cover on farms and the forms of agroforestry that was observed, however, varied significantly amongst responder groups ($p < 0.001$). On flat ground in S1 and S2, there was a tendency of growing tree cover. On farms, olive trees were by far the most common species, with the exception of irrigated land, where fruit trees predominated.

On the farms of the interviewees, a total of 19 domesticated trees and 17 wild species were found to be present. These species were grouped by the interviewees into three categories: cultivated trees suitable for rain-fed farming, cultivated trees needing regular irrigation, and wild or natural trees on farms.

There were no appreciable regional differences in agricultural methods, plant origin, intercropping, plowing, usage of pesticides, type of fertilizer, or irrigation system (Table 2). Only 7% of farmers in S1 who practice agroforestry use only mineral fertilizer, while 3% rely solely on organic waste left on the soil after harvest. Of the 30 farmers who practice agroforestry, 70% use only organic fertilizer. These results showed the dominance of synthetic and chemical fertilizers, which are the principal tools for intensive agriculture. Because of scarcity of water, only 40% of interviewed farmers in S1 use a supplemental irrigation using traditional method mainly for vegetables. The rest (60%) rely on rain as their source of water.

Farmers produce annual crops (primarily faba beans and barley) using seeds that are just 50% of certified origin between rows of trees (mostly olives). The remaining half was either purchased locally or was leftover from the previous year's harvest. The field is plowed annually before sowing using a conventional animal-drawn plow because of the topography's (sloppy land) characteristics (80%).

3.2.2. Agricultural Practices—Vegetable Crops

Between regions, there were no discernible differences in agricultural techniques, seed origin, intercropping/plant association, speed and type of plowing, use of pesticides, type of fertilization, or irrigation system (Table 3). Different vegetable crop species were often rotated every two to three years; however, plants association practices were uncommon. In S2 compared to S1, more farmers interviewed cultivate vegetable crops (53% vs. 17% in S2 and S1, respectively). The majority of vegetables grown in S1 are used by the farmer. The farmers primarily purchase their vegetable seeds from the local market (80% in S1 vs. 66% in S2) for their crops.

However, in S2, farmers are more focused on the market (local and nearby markets), which explains the usage of certified seeds (33 percent use them compared to 20 percent in S1) and higher use of mineral fertilizer (33 percent in S2 compared to 0 percent in S1). Additionally, farmers used more chemicals (94% in S2 vs. 40% in S1), including pesticides to protect their crops from diseases and pests, and drip irrigation systems had a superior water use efficiency (33% vs. 20% in S2 and S1, respectively).

At both locations, all of the farmers plow their fields every year. Further, approximately 66% of farmers in S2 use modern mechanical plowing every year due to the two sites' different topographies, with S1 having steeper slopes. All farmers surveyed in S1 prepare the land with traditional animal traction plowing. Farmers in S1 produce a wider range of vegetables (40% compared to only 5% in S2) thanks to the use of plant associations. Each and every farmer surveyed rotates the crops grown on their property.

### 3.2.3. Agricultural Practices—Cereals and Legumes

Intercropping/plant association and rate of plowing were two agricultural practices factors that varied significantly ($p < 0.05$) (Table 4) between locations. Origin of seeds, rotation strategy, chemical use, type of irrigation, and type of irrigation variables did not significantly differ from one another (Table 4). In S1 and S2 sites, respectively, 70% and 55% of farmers questioned the use of certified seeds for grains and legumes.

The majority of farmers in S1 (83.3%) adopted a two-year rotation of cereals and legumes more frequently than farmers in S2 (30%). For species of cereals and legumes that are mostly rain-fed, neither the farmers' comments nor the survey results indicated any associations with other crops. Every year, farmers on both sides plow their fields in order to prepare the soil for planting. Both sites had a higher percentage of mineral fertilization (69% vs. 55% in S1 and S2, respectively). Depending on the availability, farmers will occasionally use the dung from their livestock for fertilizer. In both locations, more than 75% of farmers use chemicals to safeguard their crops.

### 3.3. Analysis of Agricultural Production System regarding Agroecology Basis

Our findings, however, present regional practices while capturing regional heterogeneity in farm characteristics and farming systems at the small landscape scale. Rapid evaluation of the agroecological conditions and agroforestry techniques present at our study locations was made possible by local knowledge research. Here, the emphasis will be on contrasting the above-mentioned local techniques with those advocated by agroecology in order to achieve effective soil management.

**Table 2.** Agricultural practices—Agroforestry-.

| Question | Variables | Zerhoun N= | | Middle Atlas N= | |
|---|---|---|---|---|---|
| | | n | % | N | % |
| Origin of plants $X^2 = 7.529$ | Certified plants | 15 | 50.0 | 10 | 29.4 |
| | Uncertified plants | 15 | 50.0 | 14 | 41.2 |
| Intercropping/ Plants association $X^2 = 5.821$ | Yes | 21 | 70.0 | 7 | 20.6 |
| | No | 9 | 30.0 | 17 | 50.0 |
| Pace of plowing $X^2 = 7.980$ | Annually | 27 | 90.0 | 17 | 50.0 |
| | Not annually | 3 | 10.0 | 5 | 14.7 |
| | Any plowing | 0 | 0 | 2 | 5.9 |
| Type of plowing $X^2 = 8.832$ | Traditional | 24 | 80.0 | 8 | 23.5 |
| | Modern | 5 | 16.7 | 10 | 29.4 |
| | Any plowing | 1 | 3.3 | 6 | 17.6 |
| Use of chemicals $X^2 = 5.546$ | Yes | 5 | 16.7 | 16 | 47.1 |
| | No | 25 | 83.3 | 8 | 23.5 |
| Fertilization $X^2 = 17.885$ | Organic | 21 | 70.0 | 7 | 20.6 |
| | Mineral | 2 | 6.7 | 7 | 20.6 |
| | Both | 6 | 20.0 | 10 | 29.4 |
| | Any fertilization | 1 | 3.3 | 0 | 0 |
| Irrigation $X^2 = 11.276$ | Drip irrigation | 0 | 0 | 10 | 29.4 |
| | Traditional irrigation | 12 | 40.0 | 11 | 32.4 |
| | Any irrigation | 18 | 60.0 | 3 | 8.8 |
| Tree pruning $X^2 = 9.180$ | Annually | 20 | 66.7 | 22 | 64.7 |
| | Any pruning | 1 | 3.3 | 0 | 0 |
| | Rarely | 9 | 30.0 | 2 | 5.9 |

### 3.3.1. Soil Tillage and Conservation Management

Since it is done annually by all of the interviewees, soil tillage and plowing are regarded by farmers in both sites as a necessary procedure before sowing. However, there is one exception: only 71% of farmers in S2 (Middle Atlas) plow their land annually for agroforestry. Modern or conventional tillage equipment causes the earth to be turned, exposing the soil layer to evaporation and the fauna to unfavorable environmental conditions. Farmers in these locations are unaware of conservation tillage, which preserves the physical, chemical, and biological characteristics of the soil [32]. Further, sometimes, tillage in the slope direction has been noticed and leads imminently to soil loss through water erosion. This was captured in agroforestry and cereal/legumes production systems.

**Table 3.** Agricultural practices-Vegetable crops (* denote significant difference, * equivalent to $p < 0.05$, ** equivalent to $p < 0.001$, and *** equivalent to $p < 0.0001$).

| Question | Variables | Zerhoun N= | | Middle Atlas N= | |
|---|---|---|---|---|---|
| | | n | % | N | % |
| Origin of seeds $X^2 = 23.205$ ** | Certified plants | 1 | 3.3 | 6 | 17.6 |
| | Uncertified plants | 4 | 13.3 | 12 | 35.3 |
| Intercropping/ Plants association $X^2 = 21.778$ ** | Yes | 2 | 6.7 | 1 | 2.9 |
| | No | 3 | 10.0 | 17 | 50.0 |
| Rotation system $X^2 = 4.458$ | Yes | 5 | 16.7 | 18 | 52.9 |
| | No | 0 | 0 | 0 | 0 |
| Pace of plowing $X^2 = 4.458$ | Annually | 5 | 16.7 | 18 | 52.9 |
| | Not annually | 0 | 0 | 0 | 0 |
| | Any plowing | 0 | 0 | 0 | 0 |
| Type of plowing $X^2 = 9.973$ * | Traditional | 5 | 16.7 | 6 | 17.6 |
| | Modern | 0 | 0 | 12 | 35.3 |
| | Any plowing | 0 | 0 | 0 | 0 |
| Use of chemicals $X^2 = 21.84$ ** | Yes | 2 | 6.7 | 17 | 50.0 |
| | No | 3 | 10.0 | 1 | 2.9 |
| Fertilization $X^2 = 29.801$ *** | Organic | 4 | 13.3 | 2 | 5.9 |
| | Mineral | 0 | 0 | 6 | 17.6 |
| | Both | 1 | 3.3 | 10 | 29.4 |
| | Any fertilization | 0 | 0 | | |
| Irrigation $X^2 = 13.203$ * | Drip irrigation | 1 | 3.3 | 6 | 17.6 |
| | Traditional irrigation | 4 | 13.3 | 12 | 35.3 |
| | Any irrigation | 0 | 0 | 0 | 0 |

### 3.3.2. Crop Management

Based more on the farmer's finances than on preserving native genotypes, a combination of certified and local seeds is employed. Almost all farmers, especially in S1, who participated in the poll stated they rotate cereals and legumes every two years. However, when growing vegetables for the family's use, the rotation is primarily determined by the seeds that are available and by the family's needs. The survey revealed that associations between plants and vegetables were made based on ancestors' knowledge and the avail-

ability of seeds rather than on the characteristics of the plant's above- and below-ground structures (types of leaves and root system, rooting depth, species compatibility, etc.).

**Table 4.** Agricultural practices—cereals and legumes.

| Question | Variables | Zerhoun N= | | Middle Atlas N= | |
|---|---|---|---|---|---|
| | | n | % | N | % |
| Origin of seeds $X^2 = 4.764$ | Certified | 18 | 60.0 | 11 | 32.4 |
| | Uncertified | 8 | 26.7 | 9 | 26.5 |
| Intercropping/ plants association $X^2 = 0.336$ | Yes | | | | |
| | No | 0 | 0 | 0 | 0 |
| | | 26 | 86.7 | 20 | 58.8 |
| Rotation system $X^2 = 6.476$ | Yes | | | | |
| | No | 25 | 83.3 | 6 | 17.6 |
| | | 1 | 3.3 | 14 | 41.2 |
| Pace of plowing $X^2 = 0.336$ | Annually | 26 | 86.7 | 20 | 58.8 |
| | Not annually | 0 | 0 | 0 | 0 |
| | Any plowing | 0 | 0 | 0 | 0 |
| | | 17 | 56.7 | 2 | 5.9 |
| Type of plowing $X^2 = 2.342$ | Traditional | 9 | 30.0 | 18 | 52.9 |
| | Modern | 0 | 0 | 0 | 0 |
| | Any plowing | | | | |
| | | 23 | 76.7 | 15 | 44.1 |
| Use of chemicals $X^2 = 2.994$ | Yes | 3 | 10.0 | 5 | 14.7 |
| | No | | | | |
| | | 1 | 3.3 | 2 | 5.9 |
| Fertilization $X^2 = 17.428$ | Organic | 18 | 60.0 | 11 | 32.4 |
| | Mineral | 5 | 16.7 | 5 | 14.7 |
| | Both | 2 | 6.7 | 2 | 5.9 |
| | Any fertilization | | | | |
| Type of irrigation $X^2 = 4.973$ | Drip irrigation | 0 | 0 | 2 | 5.9 |
| | Traditional irrigation | 1 | 3.3 | 5 | 14.7 |
| | Any irrigation | 25 | 83.3 | 13 | 38.2 |

### 3.3.3. Nutrient Management

Although these types of studies are subsidized by the Ministry of Agriculture (subsidies may cover up to 100% of the cost for small farmers), farmers in both locations do not use them to gain an estimate of the fertility level of their soil. Both regions' cereal farmers are accustomed to using mineral fertilizer during plow-time. However, the amount used is fixed at random and again depends on the farmer's finances. However, farmers do not use fertilizer on legumes because they think, based on ancestral knowledge, that these plants can meet their own needs for mineral nourishment. As a result, they do not fertilize succeeding crops with nitrogen. Vegetable crops utilize the application of organic matter as an organic fertilizer and are dependent on the availability of manure, a byproduct of their animals. Again, the amount of manure sprayed is solely dependent on availability, which in turn depends on the number of animals the farmer has, not the needs of the crop. Farmers use all the biomass they create; nothing is left over for the soil layers to incorporate. Consequently, the current agricultural production technique depletes the soil over time.

### 3.3.4. Water Management

Farmers in both areas are aware of the water shortage and the decline in water they receive annually over time. To produce vegetables that are more profitable for them, the water that is available is used more effectively. Vegetable cultivation parcels are spread out around the center of the settlements on terraces near water sources. Drip irrigation systems were also constructed for more effective use of water when it was practicable (enough water and funds were available). Trees and crops in agroforestry rely solely on rainfall; irrigation is not used. For rainwater collection, water conservation techniques are used around trees like impluvium.

### 3.3.5. Biodiversity Management

The contextual knowledge, perceptions, and inspirations of the targeted farmers were not sufficiently considered during the phases of design, implementation, and evaluation of development processes, which further contributed to the slow or incomplete adoption of the agroforestry innovations of recent decades. On the other hand, natural hedges are a practice increasing biodiversity [33].

### 3.4. Farms Characteristics and Crop Types/Agricultural Practices Relationship

Figure 2 is an ordination diagram that shows the link between the features of farms and crop types. The first two canonical axes are in the horizontal and vertical directions, respectively, and the arrows stand for the various characteristics of farms' variables. The length of the arrows shows the relative contribution of the variables to the axes and the features of the farms, while the direction of the arrows represents the correlation between each variable and the canonical axes, and each other. Relationships between crop kinds and the features of participating farmers are also represented.

In general, the first two canonical axes account for an estimated 89.23% of the diversity in the characteristic-crop kinds, whereas the first axis accounts for an estimated 50.92%. The choice of crop varieties was considerably influenced by region and organization type, but not by geography or the size of the farmland. The organization and topography arrows in Figure 2's different directions showed that these two elements had opposing effects on the sorts of crops. Therefore, area and terrain had a significant role in advancing knowledge of the choosing of crop types. As a result, terrain and area played a significant role in promoting a knowledge of crop type selection. Region and organization type were crucial determinants of agroforestry techniques, but topography and agricultural area were not; the first two axes account for an estimated 93.81% of the total. According to estimates, the first canonical axes account for 59.09% of the diversity in the farms' features and agricultural practices link between vegetable crops, while the first two axes account for 83.09%. The arrows in Figure 2 for region and organizational type pointed in opposite directions, indicating that these two factors had opposing influences on the majority of agricultural practices of vegetable crops. Region and farmland area were significant factors that affected agricultural practices of vegetable crops. The first two canonical axes comprise an estimated 81.78% of the variation in the association between farm characteristics and cereal and legume agricultural methods, with the first axis accounting for an estimated 58.55% of that variation.

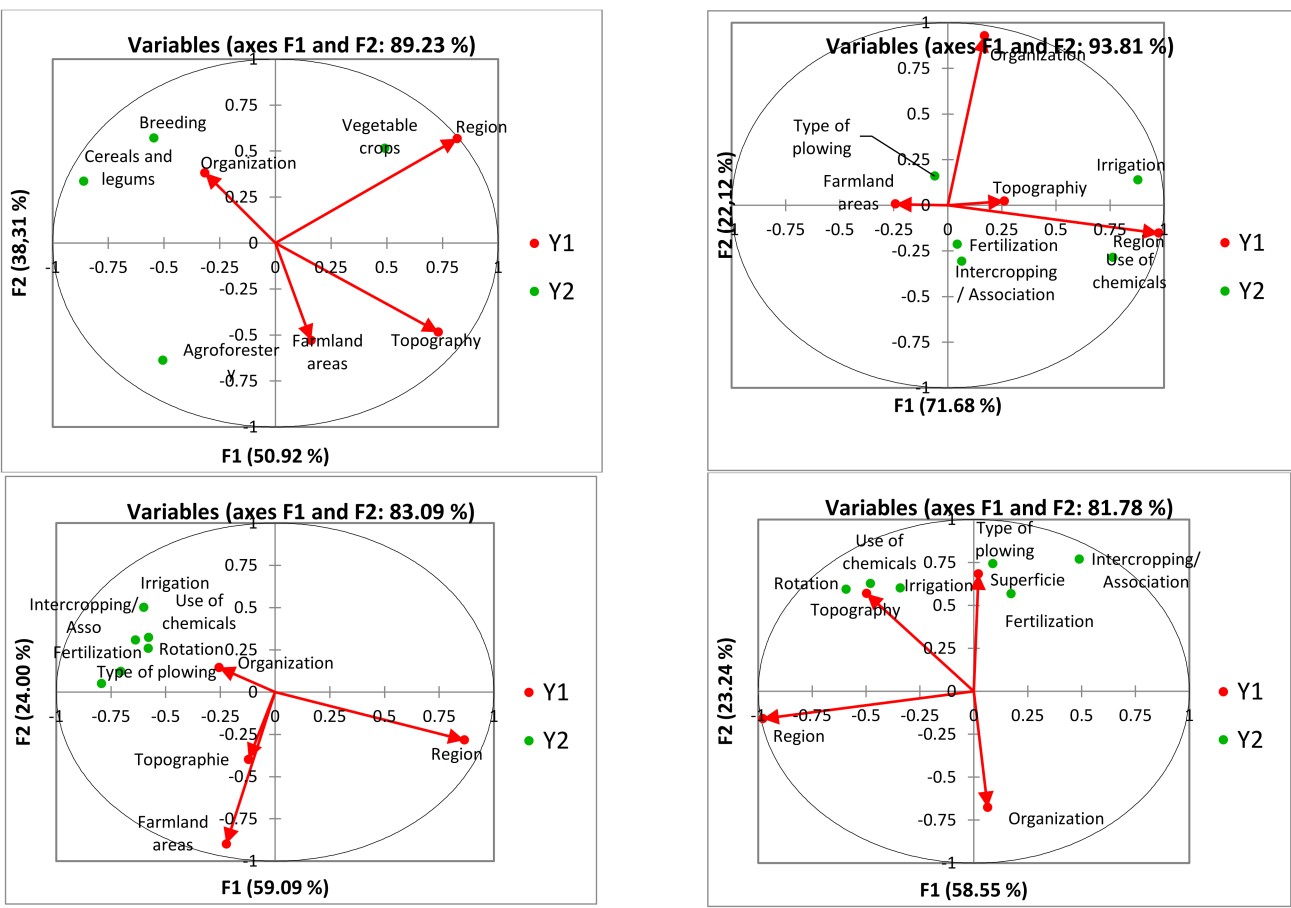

**Figure 2.** Characteristics of farms, crop types and agricultural practices of interviewees under CCA constructed according to data collected. Full questions are presented in Tables 1 and 2.

## 4. Discussion

This study highlighted for the first time the agricultural production systems and farming practices in Moulay Driss Zerhoun and the Middle Atlas in central Morocco. Our results defined the characteristics of farmers and described the farm production systems, including agricultural practices such as agroforestry and vegetable crops. Equally, we analyzed the farm production system in the study area regarding agroecology basis, counting soil tillage and management of crops, water, biodiversity, and conservation, which are among the missing pieces of data on Moroccan agro-ecosystems.

During four years of the census, S2 had more interviews with farmers than S1 and only male farmers were contacted. In S1, all participants are farmers and were located in Moulay Driss Zerhoun's urban region. The majority of farms at both locations are between 0 and 5 ha, S2 generally has a larger farm size than S1, with farms larger than 5 ha. Nonetheless, farms in both locations are tiny in size and have numerous parcels in addition to that. Similar results were recorded by Faysse, who reported that the majority of Moroccan farmers have small-scale farms [34]. In our situation, the majority of the farms were located wholly or in part on muddy terrains due to the disparities in topography between the two locations, while a minority were located on flat land in S2 and S1. Mensour et al. [35] noted that the farmlands in the Souss-Massa region of southern Morocco were situated on flat terrains, which is consistent with our findings. The farms that were examined in this study fall under the international classification's category for rain-fed mixed systems. Additionally, the farming practices in northern Morocco have long been referred to as mixed rain-fed systems [15]. According to El-Shater and Yigezu [36], Lemerle [37], and Thornton [38], irrigated farms matched the description of small-scale irrigation subsystems, whereas



lower slope, mountain farm and livestock owners had characteristics corresponding to both rain-fed mixed systems and highland mixed systems [39].

Agroforestry is often practiced in both zones. Olive, carob, fig, and caper shrubs make up the majority of S1, while fruit trees make up the majority of S2 (apples, cherries, peaches, prunes, pears, etc.). On the Saiss plain, similar consequences are currently being seen [15,40]. By contrast, in our situation, only a few farmers in S1 plant vegetables for their own consumption. The linkages between olive trees and market gardening, olive trees and fodder, and olive trees and annual crops were noted, which is similar to the findings published for the West Anti-Atlas (Morocco) agroforestry systems [41]. The three agricultural production techniques used by farmers in both sites are agroforestry, vegetable production, and grain and legume production, which is consistent with earlier findings mentioned in the Saiss Plain (Morocco), known for its sustainability and diversified farming methods that produce vegetables, cereals, and legumes [42,43]. This high productivity is made possible by the abundance of flat lands, fertile soil, and water, all of which are essential for agriculture and agroforestry [44,45].

On the flat ground at the tested sites, there was a propensity for tree cover to grow. Except for irrigated areas, where fruit trees predominated, olive trees were by far the most prevalent species on farms. The Agroforestry Systems of the Saiss Region, which is close to our study sites, found similar outcomes [22]. Additionally, among cultivated plants, a total of 19 domesticated trees and 17 wild species were identified. These planted species were divided into three groups: farm trees that could be grown using only rainwater, farm trees that required regular watering, and farm trees that were grown naturally. Along with oranges, vines, figs, apples, and peaches, as well as wild species like *Tamaris sp.*, *Phragmites australis*, and *Typha angustifolia*, Kouchou et al. [18], Jabiri et al. [46], and Squalli et al. [47,48] described olives as prominent trees. Only a small percentage of farmers in S1 used only mineral fertilizer, whereas the majority of farmers used organic fertilizer. These findings demonstrated the domination of synthetic and chemical fertilizers, which were described by Kouchou et al. [18] in the same region as the main instruments for intensive agriculture. This domination risk to pollute soil and groundwater with chemical substances. Farmers utilize supplemental irrigation based on conventional techniques mostly for the irrigation of vegetables due to the lack of water in the study zone [49]. In addition, some farmers only get their water from rain.

In the research locations, different vegetable crop species were frequently rotated every two to three years, although plant association activities were rare. According to the findings stated in the same region, the farmers use the majority of the vegetables grown in S1 and typically buy their vegetable seeds from the local market for their harvests [50]. At both sites, all of the farmers plow their fields annually. However, in S2, due to the topography noted by Dauteuil et al. [51], farmers employ modern mechanical plowing every year. All farmers in S1 employ traditional animal traction plowing to prepare the soil, and by utilizing plant associations, they can grow a broader variety of vegetables. Additionally, farmers use certified seeds for grains and legumes, which is currently noted by El Ansari et al. [43]. Rotation strategy, chemical use, type of irrigation, and type of irrigation variables did not significantly differ from one another. In our situation, the higher percentage in S1 compared to S2 can be attributed to the fact that dealers of certified seeds are simpler to locate close to Moulay Driss Zerhoun town. In addition, more S1 farmers than S2 farmers used a two-year rotation of cereals and legumes. Farmers plow their fields annually to prepare the ground for planting on both sides [50].

Our findings highlight regional behaviors while capturing, at the small landscape scale, regional variety in farm characteristics and farming systems. According to our research, farmers in both study sites view soil tillage and plowing as a vital step before sowing, which is consistent with observations made in the Saiss plain around our study sites [50]. In S2 (Middle Atlas), the majority of farmers plow their land each year for agroforestry. However, because the earth is moved by contemporary tillage machinery, the soil layer is exposed to evaporation and the fauna is subjected to unfavorable environmental conditions.

In our situation, farmers in these areas are not aware of conservation tillage, which protects the biological, chemical, and physical properties of the soil and was previously noted in the same region by Daoui and Fatemi [22]. Every two years, the farmers under investigation switch out their cereals and legumes. The cycle, however, is mostly controlled by the family's needs and by the seeds that are accessible for growing vegetables for the family's consumption. Instead of taking into account the properties of the plant's above- and below-ground structures, the linkages between plants and vegetables were created based on the knowledge of their ancestors and the accessibility of seeds. According to current results in the same locations, cereal farmers in both regions are accustomed to utilizing mineral fertilizer throughout the plow season [18]. The quantity used, however, is fixed arbitrarily and again is based on the farmer's financial situation. Because they believe, based on ancient knowledge, that these plants are capable of meeting their own needs for mineral feeding, farmers do not use fertilizer on legumes [52]. As a result, they do not apply nitrogen fertilizer to succeeding crops. The quantity of manure sprayed is completely based on availability, which in turn depends on how many animals the farmer has, not on the crop's demands. Farmers in both regions are aware of the water crisis and the gradual reduction in the amount of water they receive each year. As a result, researchers looked at how well farmers were using the water that was available to grow vegetables that would be more profitable for them [45,53]. In our situation, vegetable farming portions were dispersed across the settlement's center on terraces near water sources. When it was feasible, drip irrigation systems were also built for more efficient use of water (enough water and funds were available). The use of herbicides and pesticides against weeds, pests, and illnesses will undoubtedly have an impact on the biodiversity of above-ground (plants) and below-ground fauna and flora, with the exception of local vegetation that still remains in part on borders around farmers' parcels [18,54]. Chemicals have been shown to interfere with several natural processes, including the bacterial breakdown of organic waste and earthworm-mediated changes in soil structure and porosity.

## 5. Conclusions

This study presents the first investigation of agricultural production systems and agricultural practices in Moulay Driss Zerhoun and the Middle Atlas in central Morocco. Our principal objective was to evaluate the gap between local agricultural practices and agroecology principles and recommendations. We demonstrated that the majority of farmers were rural and have small farms of 0 to 5 ha. Both zones were dominated by agroforestry of arboriculture, in addition to annual crops. Olive trees were the most dominant and planted in association with fodder and market gardening. In total, 19 domesticated trees and 17 wild species were found in these sites. The majority of farmers use organic fertilizers and rely on rain for irrigation. Half of farmers of annual crops use certified seeds, while the rest purchase locally or use what was left over from the previous year's harvest. To safeguard their crops, farmers use chemicals. Before cultivation, soil tillage and plowing are regarded as necessary procedures before sowing, and, only in S2, farmers plow their land annually for agroforestry. The choice of crop varieties was considerably influenced by region and organization type, but not by geography or the size of the farmland, and area played a significant role in promoting knowledge of crop type selection. The diversity in the farms' features and agricultural practices link to vegetable crops. These findings are the first in this study area and demonstrated that the agricultural systems are far from agroecology principles. Despite the importance of this study in evaluation of the interval between agroecology principles and local practices in study areas, as well as for comparative investigations, more advanced studies are needed to compete the gap and offer necessary data for decision makers. The future studies must investigate the impacts of local practices on quality of soil, groundwater and produced cultures.

**Author Contributions:** Conceptualization, A.I. and L.N.; methodology, validation, and formal analysis, A.I., F.R., M.E.J., R.L. and L.N.; writing—original draft preparation, A.I.; writing—review and editing, R.L., M.E.J., and F.R. visualization, F.R. and G.E. All authors have read and agreed to the published version of the manuscript.

**Funding:** This research received no external funding.

**Informed Consent Statement:** Not applicable.

**Data Availability Statement:** The data used to support the findings of this study are included within the article.

**Conflicts of Interest:** The authors declare no conflict of interest.

## Appendix A

**Table A1.** The questionnaire on farms identification and crop types.

| Questions |
| --- |
| Part 1: identification of farms |
| 1. Commune/Organization |
| 2. Farmland areas/Topography of farms/methods of fences |
| Part.2: Crop types |
| 1. Agroforestry |
| 2. Vegetable crops |
| 3. Cereals/legumes |
| 4. Breeding |

**Table A2.** The questionnaire on farmer's practices.

| Questions |
| --- |
| **Part 1: Agricultural practices of Agroforestry** |
| 1. What is the origin of the plants? |
| 2. Do you opt for intercropping or crop association? |
| 3. What is the pace of plowing? |
| 4. Do you use chemical products? |
| 5. Do you opt for organic or mineral fertilization? |
| 6. What is the rate of tree pruning? |
| 7. What type of irrigation you use? |
| **Part 2: Agricultural practices of vegetable crops** |
| 1. What is the origin of seeds? |
| 2. Do you opt for intercropping or crop association? |
| 3. Do you opt for the practice of rotation? |
| 4. What is the pace of plowing? |
| 5. You use chemical products? |
| 6. You opt for organic or mineral fertilization? |
| 7. What type of irrigation you use? |
| **Part 3: Agricultural practices of cereals and legumes** |
| 1. What is the origin of seeds? |
| 2. Do you opt for intercropping or crop association? |
| 3. Do you opt for the practice of rotation? |
| 4. What is the pace of plowing? |
| 5. You use chemical products? |
| 6. You opt for organic or mineral fertilization? |
| 7. What type of irrigation you use? |
| **Part 4: type of breeding** |
| 1. Type of breeding |
| 2. Manure destination |

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
