# Peer review of "Description of the Gap between Local Agricultural Practices and Agroecological Soil Management Tools in Zerhoun and in the Middle Atlas Areas of Morocco"

_land, doi:10.3390/land12020268_

Round 1
Reviewer 1 Report
The study titled "Description of the gap between local agricultural practices and agroecological soil management tools in Zerhoun and in Middle Atlas areas of Morocco" is assessing the farmers' knowledge gap between local agricultural practices and agroecological principles and practices. This study therefore has potential to conribute in the adoption of farming practices that will promote soil fertility and soil health and mitigation of climate change through agroecological farming practices. I recommend the following for improvement of the quality of the manuscript:
Abstract: State clearly the background or rationale of this study with reference to the study area.
Introduction:
1. Author (s) must start sentence in Line 41 to Line 43 with "Morocco's population will be particularly susceptible to food insecurity in the future due to the country's growing..."
2. The aim of the study is not captured sufficiently and I request author(s) to look at the sentence in Abstract in Line 15 to Line 18 and then rewrite the aim of the study in Line 67 to Line 69.
3. Sentence starting with "This agricultural Plain... in Line 70 to Line 72 must be moved to Study area under Materials and Methods.
4. Rephrase study objective (i) in Line 72
5. State the hypothesis of the study as it is currently missing.
Materials and Methods
6. Include coordinates about location of the study areas where the surveys were carried out within these two regions.
7. Under sub-section Data analysis, author(s) need to specify the SPSS version that was used and SPSS must be written in full as it is used for the first time in the manuscript.
Results
8. Remove "in addition to that" in sentence under Line 145 to Line 146.
9. Indicate level of significance with P-value in sentence under Line 185 to Line 188 starting with "The extent of tree cover..."
10. Indicate level of significance with P-value in sentence under Line 235 to Line 236.
11. On Page 8, Table 5, indicate what does each asterisk symbol mean e.g. *; **; and ***. This will help readers to easily understand your Table information.
12. In sentence under Line 268 to Line 269 indicate the percentage of farmers who stated that they rotate creals and legumes, as crop rotation is one of the important aspects of agroecological practices.
13. Sentence starting with "Relationship between..." under Line 309 is incomplete and rewrite this sentence.
14. Indicate Figure number in sentence starting with "The arrows..." under Line 322 as the figure number is currently missing.
Discussions
15. In Line 383 with sentence starting with "These findings demonstrated the domination of synthetic and chemical fertilizer..." you need to indicate the implication of this domination.
Conclusions
16. Author(s) need to indicate clearly what conclusion can be drawn from the findings of this study with reference to knowledge gaps between local agricultural practices and agroecological practices. This is lacking in the study in its current form.
17. Author(s) need to include the section on Recommendations and Limitations in this study. This section on Recommendations is important because the recommendations are part of the key objectives of the study (see last paragraph of Introduction).
Tables
18. Some Tables lack statistical analysis results yet in the text statements like "varied significantly" often appear. This must be indicated clearly in the Table and the meaning of symbols like *; **; *** must be clearly stated including p-values.
Author Response
See file attached.

Reviewer 2 Report
Line 1- 36. The abstract appears not to address the critical issue in the topic: the gap between agricultural practices and agroecological soil management. The description is one-sided, mainly describing what the local farmers are doing. What is the standard on the other side? the agroecological soil management practices? What is the gap? is it a convergence between local agricultural practices on the one hand and agroecological soil management on the other? Is it a productivity (output/unit input) gap? or is it a farm/production cost gap? As it stands, the topic promises more than it delivers.
41-43 should read. Morocco may not be to deal with the challenges of food insecurity in the future due to the growing population and rise in food demand, coupled with weak adaptive capacities to climate variability.
50.. should read "the agricultural sector in Morocco.
52-53... should read.... with the dominant traditional agricultural system which is subsistence-oriented, largely rain-fed and employs mostly traditional farming practices.
Line 59-61..paragraph is not clear. If it is intended as the justification of the paper then it should be rephrased or removed. " There is a need for? research that adopts a (what?) perspective of analyzing local agricultural knowledge and scientific understanding of socio-ecological systems and processes of change. Research that can better identify intervention points and options tailored to specific livelihoods and agroecosystems [13]. unclear
62-66 is another misplaced paragraph. Either edit to improve flow or delete
67.. may read this paper investigates..
72-75.. there should be a mention of these objectives in the abstract.....
112.... "questionnaire was created and assembled" consider revision.
114- 115 may read "The primary objective of the survey was to collect data on the agricultural production systems in both sites and to catalogue the agricultural production practices associated with each system.
lines 120-124 should be moved to an appendix. Inserting the questionnaire in the middle of a write-up is unconventional.
paper is silent on how the sampling of farmers/farms was done.
124.. may read raw data was.... coded and recorded into excel..
140.... In neither of the locations, renting an agricultural land is typical... rephrase. may consider "Renting of agricultural land was not common in both study sites..."
141-142.... however, 142 occasionally a widespread practice in the region... is redundant.
149 just 13% consider and only 13%
154-155 revise. consider "fruit trees including olive, fig, carob, and almond trees; cereals, legumes, forages and vegetables".
Author Response
See file attached

Reviewer 3 Report
This study aims to assess the gap between local agricultural practices and the principles and recommendations of agroecology in Central Morocco. There are some comments that are recommended to be addressed prior to consideration of publication.
1. In the abstract please refer to the methodology that you used in order to analyze the research data (Descriptive statistics? Chi-square?).
2. Regarding the introduction section please quote a research background, research significance, purpose, and contribution. Present a literature gap and make clear why the literature needs your study in the field of agriculture in general, production systems, soil tillage, and conservation management.
3. It is usually recommended to refer to the paper’s structure at the end of the introduction section.
4. Is there any literature source that could support the statement: “This agricultural plain is considered one of the most fertile and productive regions in Morocco, as well as this plain, is dominated by small farms.”?
5. Please add the source for figure 1. Even if it is an own processing figure it is recommended to refer it too. Please do the same for figure 2.
6. Is the questionnaire based on other literature studies? If yes please cite them in section 2.2. If not you need to support the questionnaire by reviewing the relative literature.
7. As I understood, the interviews were personal. Who made these interviews? Was she/he trained on this specific issue in order to make them? If yes you could refer that the interviews made by the authors that were previously trained on this specific issue/field.
8. Please support with the help of evidence from literature why did you select to analyze the data by using the chi-square test, canonical correspondence analysis, etc. Did you support this choice from another relative study? If yes please add these studies in section 2.3.? It is recommended to do the same for the variables’ selection.
9. Usually, in the section of materials and methods, the methodologies used are described. Please make a corresponding addition through a new sub-section and describe, in detail, the methods that you used in order to analyze the data.
10. Nicolas Faysse (2015), Mensour et al. (2019), El-Shater and Yigezu (2010), Lemerle (2011), Thornton (2018) etc. Please follow the journal’s rules on how you have to cite the sources throughout the paper.
11. In the Conclusions section should be, in detail, referred the paper’s contribution to the literature, farmers, agriculture, economy, etc.
Author Response
See file attached.

Round 2
Reviewer 1 Report
The author(s) have addressed my comments.
Reviewer 3 Report
No comment